# Post-Calibration Techniques: Balancing Calibration and Score Distribution Alignment

**Agathe Fernandes Machado**
Université du Québec à Montréal
201 Av. du Président-Kennedy, Montréal, QC H2X 3Y7, Canada
`fernandes_machado.agathe@courrier.uqam.ca`

**Arthur Charpentier**
Université du Québec à Montréal
201 Av. du Président-Kennedy, Montréal, QC H2X 3Y7, Canada
`charpentier.arthur@uqam.ca`

**Emmanuel Flachaire**
Aix-Marseille School of Economics, Aix-Marseille Univ.
5 Bd Maurice Bourdet CS 50498, 13205 Marseille Cedex 01, France
`Emmanuel.flachaire@univ-amu.fr`

**Ewen Gallic**
Aix-Marseille School of Economics, Aix-Marseille Univ.
5 Bd Maurice Bourdet CS 50498, 13205 Marseille Cedex 01, France
`ewen.gallic@gmail.com`

**François Hu**
Milliman France
14 Av. de la Grande Armée, 75017 Paris, France
`hu.faugon@gmail.com`

## Abstract

A binary scoring classifier can appear well-calibrated according to standard calibration metrics, even when the distribution of scores does not align with the distribution of the true events. In this paper, we investigate the impact of post-processing calibration on the score distribution (sometimes named "recalibration"). Using simulated data, where the true probability is known, followed by real-world datasets with prior knowledge on event distributions, we compare the performance of an XGBoost model before and after applying calibration techniques. The results show that while applying methods such as Platt scaling, Beta calibration, or isotonic regression can improve the model's calibration, they may also lead to an increase in the divergence between the score distribution and the underlying event probability distribution.

## 1 Introduction

When estimating a probabilistic scoring classifier, the model must not only discriminate between observations according to their class but also return scores that can be interpreted as probabilities. The

Workshop on Bayesian Decision-making and Uncertainty, 38th Conference on Neural Information Processing Systems (NeurIPS 2024).

distribution of scores produced by the classifier should align with the underlying event distribution. To assess whether classifiers return probabilistic scores, one must evaluate the model's calibration [4, 24, 8]. While some models, such as logistic regression when correctly specified, are known to be well-calibrated [22], others, including ensemble methods like Random Forests (RF) [13, 3] and XGBoost [10], are not inherently calibrated [28]. To assess a model's ability to provide probabilistic scores, the literature recommends evaluating its calibration using metrics like the Brier Score (BS, [4]) or the Integrated Calibration Index (ICI, [1]). When a model is not well-calibrated, post-processing calibration methods including Platt scaling [29], Beta calibration [17], or isotonic regression [37] are often applied to adjust the scores [21, 11, 17]. After applying calibration techniques, these metrics generally indicate an improvement in the model's calibration relative to its initial state.

Since the true underlying probability distribution of the data is typically unobserved in practice, calibration metrics are assessed solely on the classifier's output range. Fernandes Machado et al. [9] demonstrated with simulated data that methods such as RF and XGBoost, can appear well-calibrated according to standard calibration metrics and exhibit strong discrimination based on performance metrics, yet still fail to align the score distribution with the true event distribution. This discrepancy can arise when predicted scores from those algorithms lack the heterogeneity present in the underlying data distribution. They demonstrate this misalignment by comparing the selection of model hyperparameters based on Kullback-Leibler (KL) divergence with the selection based on performance or calibration metrics, knowing the true event distribution in the case of simulated data. For real data, where the true distribution is unknown, the approach involves prior information about the underlying data distribution to better align it with the predicted score distribution. Their analysis only considers model evaluation to accurately interpret predicted scores as probabilities, typically through calibration metrics. However, many practitioners employ post-calibration techniques to ensure that output scores represent probabilistic estimates. In this paper, we examine how post-calibration techniques affect the variability of score distribution in XGBoost binary classifiers, comparing it to the true underlying data distribution using KL divergence. We find that these post-processing methods often reduce score heterogeneity. Additionally, the misalignment between tree-based models optimized for KL divergence and those optimized for calibration or performance metrics persists and may even worsen after calibration, indicating that score alignment can decrease following post-calibration. Using simulated data with known true probabilities, followed by real-world datasets with prior knowledge of event distributions, we evaluate the abilities of XGBoost-predicted scores as probabilistic estimates before and after applying calibration techniques, with an emphasis on score distribution rather than solely on their calibration.

## 2   Calibration

We focus on the context of a binary scoring classifier. Let $Y \in \mathcal{Y} = \{0, 1\}$ be a binary response variable, and let $\boldsymbol{X} \in \mathcal{X} = \mathbb{R}^d$ denote features. The goal is to predict $s(\boldsymbol{X}) = \mathbb{P}(Y = 1 | \boldsymbol{X})$, using a sample of $n$ i.i.d. observations $(\mathbf{x}_i, y_i)_{i=1}^n$. We estimate this probability $\hat{s}(\mathbf{x}_i) \in [0, 1]$ using an XGBoost classifier, which produces a distribution of estimated scores $\hat{s}(\boldsymbol{X})$. If the score distribution is poorly calibrated, these scores cannot be interpreted as the "true underlying probabilities" [33, 19, 16]. A model $\hat{s}$ is well-calibrated for a binary variable $Y$ when [31]:

$$\mathbb{P}(Y = 1 \mid \hat{s}(\boldsymbol{X})) = \mathbb{E}[Y \mid \hat{s}(\boldsymbol{X})] = \hat{s}(\boldsymbol{X}) \quad \text{a.s.}, \tag{1}$$

i.e., equivalently, $\mathbb{E}[Y \mid \hat{s}(\boldsymbol{X}) = p] = p, \forall p \in [0, 1]$.

### 2.1   Calibration Metrics

To measure calibration, the literature suggests various metrics. Here, we focus on two of them: BS and ICI. The former [12, 17, 29, 30], often used to assess a model's calibration, is a proper scoring rule that also accounts for refinement loss [18]. It writes: $\text{BS} = n^{-1} \sum_{i=1}^n \left( \hat{s}(\mathbf{x}_i) - y_i \right)^2$ [4]. More recently, Austin and Steyerberg [1] introduced the ICI, a metric that relies on the calibration curve. In the binary case, the calibration curve writes $\text{g} : [0, 1] \to [0, 1], \quad p \mapsto \text{g}(p) := \mathbb{E}[Y \mid \hat{s}(\mathbf{X}) = p]$. For a well-calibrated model, the calibration curve corresponds to the identity function, $\text{g}(p) = p$, where the predicted score $p$ equals the true likelihood of the event. Graphically, this is represented by the calibration curve aligning with the 45-degree diagonal. While the calibration curve is usually estimated using bins [35, 20, 27], the ICI relies on a smoother version, based on splines. The empirical version writes $\text{ICI} = n^{-1} \sum_{i=1}^n \left| \hat{s}(\mathbf{x}_i) - \hat{\text{g}}(\hat{s}(\mathbf{x}_i)) \right|$, which corresponds to computing the average of

the absolute difference between the estimated calibration curve and the identity function, the latter representing perfect calibration.

## 2.2 Calibration Methods

When using scores generated by a model estimating the probability of a binary event, the literature advocates calibrating the model by applying the calibration curve g—which serves as a transformation function—on the scores [29, 37, 17, 20]. In this paper, we focus on three calibration methods: Platt scaling, isotonic regression, and Beta calibration.

**Platt Scaling** This parametric approach consists of fitting a logistic regression to the binary response variable using predicted scores of a binary classifier as the unique feature [29]. The obtained calibrated probabilities are $g(\hat{s}(\mathbf{x})) = \left(1 + \exp\left\{-\frac{1}{s}(\hat{s}(\mathbf{x}) - \mu)\right\}\right)^{-1}$, where $\mu$ and $s$ ($s > 0$ for a non-decreasing calibration map g) are estimated on a calibration set. It should be noted that Platt scaling is unable to learn the identity function g if the predicted scores are already calibrated [17].

**Beta Calibration** The scores returned by a binary classifier are in range $[0, 1]$. Beta calibration [17] builds on this feature and assumes that the score within each class of the target variable $y$ are distributed according to a Beta distribution. By contrast, Platt scaling assumes the scores follow a Normal distribution within each class. The calibration map writes $g(\hat{s}(\mathbf{x})) = (1 + \exp\{-a \log \hat{s}(\mathbf{x}) + b \log(1 - \hat{s}(\mathbf{x})) - c\})^{-1}$, where $a$, $b$, and $c$ are the three parameters that need to be estimated on a calibration set. Unlike Platt scaling, Beta calibration can learn the identity function g (with $a = b = 1, c = 0$), making it suitable for already well-calibrated models. By restricting $a, b > 0$, the calibration map is monotone.

**Isotonic Regression** This solution arises from a constrained optimization problem [37], solved using the Pool-Adjacent-Violators Algorithm, ensuring that corrected predicted scores remain monotonic: $\min_{\beta_1,\ldots,\beta_n} \sum_{i=1}^{n} (y_{(i)} - \beta_i)^2$, s.t. $\beta_1 \leq \ldots \leq \beta_n$, where $y_{(i)}$ corresponds to the value in $\{y_1, \cdots, y_n\}$ associated with the $i$-th largest predicted score $\{\hat{s}(\mathbf{x}_1), \cdots, \hat{s}(\mathbf{x}_n)\}$. Isotonic regression will lead to $g(\hat{s}(\mathbf{x}_i)) = \beta_i^\star$ where $\beta_i^\star$ solve the optimization problem.

## 3 Score Heterogeneity

To accurately interpret predicted scores from a binary classifier as probabilistic estimates, since the true underlying probability $s(\boldsymbol{X})$ is usually unobservable, calibration metrics rely solely on the predicted score range. When a binary classification model is well-calibrated, the distribution of its scores $\hat{s}(\boldsymbol{X})$, as defined by Eq. 1, should align with the actual probability of the event in the vicinity of score values. Therefore, calibration metrics cannot fully capture discrepancies between the score distribution and the true probability distribution of the response variable $Y$ when the predicted score variability does not accurately reflect the latter.

**Kullback-Leibler divergence** Fernandes Machado et al. [9] demonstrated through simulated data that scores from ensemble methods may exhibit less variability compared to the true underlying probabilities when selecting hyperparameters based on calibration (ICI) or performance (AUC) metrics. This reduced heterogeneity makes calibration metrics less reliable for interpreting output scores as probabilities of event occurrence. Instead of evaluating discrepancies solely on predicted score values, the authors emphasize evaluating the model's probabilistic estimates using KL divergence between the overall score distribution, $\hat{s}(\boldsymbol{X})$, and the available information on the "true" distribution, $s(\boldsymbol{X})$. Additionally, the flexibility of tree-based methods like XGBoost enables the selection of model hyperparameters based on KL divergence instead of traditional performance metrics, ensuring the availability of a model whose predicted score distribution closely aligns with prior knowledge.

**Bayesian Framework** When working with simulated data, the distribution $s(\boldsymbol{X})$ is fully known, allowing for the direct computation of KL divergence with $\hat{s}(\boldsymbol{X})$. However, with real data, the KL divergence can only be computed by relying on a prior belief about the distribution of $s(\boldsymbol{X})$, potentially informed by expert opinion, and thus assuming a prior distribution $\mathcal{B}$. In the following, as in Fernandes Machado et al. [9], we take $s(\boldsymbol{X}) \sim \mathcal{B} = \text{Beta}(\alpha, \beta)$ as the assumed prior distribution where each probability $p_i$ of the $i$-th observation is a sample from $\mathcal{B}$. We observe a sequence of

$n$ independent (as the features $\boldsymbol{X}_i$ are considered $n$ i.i.d. random variables) but non-identically distributed binary random variables $Y_i$ where $Y_i|s(\boldsymbol{X}_i) = p_i \sim \text{Bernoulli}(p_i)$. In this case, instead of selecting the model with hyperparameters that minimize the empirical mean of the KL divergence across individual distributions, we directly minimize the distance between the prior distribution $\mathcal{B}$ and the overall distribution of $\hat{s}(\boldsymbol{X})$.

**Calibration techniques** We extend the work of Fernandes Machado et al. [9] by investigating how score heterogeneity predicted by certain XGBoost algorithms is affected after applying post-calibration techniques such as Platt scaling, Beta calibration, or isotonic regression. These methods can potentially reduce score heterogeneity; for instance, isotonic regression applies a stepwise function g. Additionally, with Platt scaling, the range of calibrated predicted scores is always narrower than the range of the initial scores when the parameter $s \geq \frac{1}{4}$ (see Appendix A.1). And, due to the concavity of the sigmoid function over $[0, +\infty]$, this post-calibration method tends to reduce the range of predicted scores more significantly when the initial scores are highly concentrated.

## 4 Numerical Experiments

### 4.1 Simulated Data

We use the simulated data from Fernandes Machado et al. [9]. We consider four data-generating processes (DGPs), all of which use a logistic link function. The first three are from Ojeda et al. [26], the fourth adds interaction terms (see Appendix B.1). For each DGP, we generate data that include more or less noise variables: 0, 10, 50 or 100. We split the data into four samples: the train and validation samples used to train an XGBoost model and select the set of hyperparameters, the calibration sample to train a calibrator using the selected model, and lastly, a test sample to assess the performance of models. We select the model's hyperparameters (number of boosting iterations and maximum tree depth) to optimize either one of three different criteria on the validation set: maximizing AUC (AUC*), minimizing KL divergence (KL*), and, for illustrative purposes, producing a model that is poorly calibrated based on the ICI metric (High ICI). Once the hyperparameters are selected, a calibration technique is applied to the scores. This allows for a comparison of models on the test set, both before and after calibration, according to the chosen optimization criterion. We run the simulations on 100 replications for each configuration. The results for DGP 1 are shown in Fig. 1 (see Fig. C17 for full results and Table C1 for numerical values). The x-axis represents calibration, measured by the ICI, where lower ICI values indicate better calibration. The y-axis shows the KL divergence between the model's predicted score distribution and the true probabilities, with lower values indicating closer alignment between the two distributions. A model is preferable when it achieves better calibration and closer alignment between score distributions and true probabilities. Shapes represent models before calibration, while arrows show their performance after applying *post-hoc* calibration. Ideally, *post-hoc* calibration improves both metrics for uncalibrated models, resulting in arrows pointing down and to the left on the graph.

When the model is selected to optimize AUC (AUC*), calibration is generally fairly good across all DGPs, regardless of the number of noise variables. Applying a *post-hoc* calibration technique typically reduces the ICI, further improving model calibration. However, Platt scaling (green solid arrows) often fails, as the logistic function lacks the identity mapping. The score distributions from models optimized for AUC, however, are poorly aligned with the true probability distributions. For noise-free datasets, the KL divergence is approximately 2.5 times larger compared to models optimized for KL divergence. This gap widens as the number of noise variables increases. When post-calibration techniques are applied to AUC-optimized models, KL divergence increases with Platt scaling and isotonic regression but decreases with Beta calibration. However, even with Beta calibration, the KL divergence remains higher than that of models optimized for KL divergence. For initially miscalibrated models (High ICI), post-calibration generally improves calibration, with improvements seemingly unaffected by the number of noise variables. However, the impact on KL divergence is more mixed, with no systematic improvement observed, particularly for DGPs 2 and 4.

Overall, while post-calibration improves model calibration, it does not consistently align score distributions with true probabilities and may even exacerbate misalignment, highlighting trade-offs between calibration and distribution alignment.

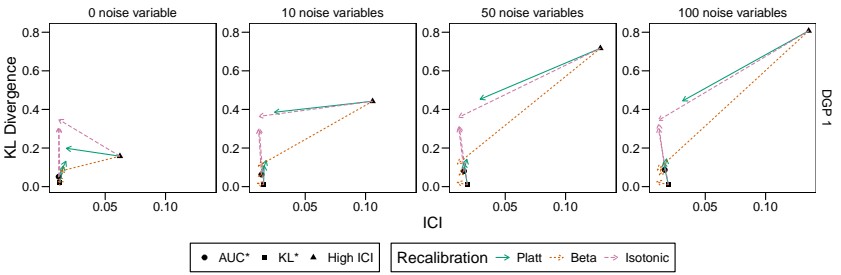

Notes: AUC*, KL*, High ICI: models selected by optimizing AUC, KL divergence, or by selecting a high ICI.

Figure 1: Average KL divergence and ICI before and after recalibration, for DGP 1.

## 4.2 Real Data

The 10 datasets from the UCI ML Repository used in Fernandes Machado et al. [9] are used here (see details in Appendix B.2). For each dataset, we apply the method outlined in Section 4.1, this time calculating the KL divergence between the predicted score distribution and the prior distribution described in Section 3.[1] The results across the 10 datasets are shown in Fig. 2, with detailed metric values in Table C2. The x-axis represents calibration with ICI, and the y-axis shows KL divergence (lower values indicate closer alignment with Beta priors). Shapes denote models before calibration, and arrows indicate changes after applying post-calibration techniques from Sec. 2.2.

The findings are consistent with Section 4.1. When applied to already calibrated scores with low ICI (models AUC* and KL*), Platt scaling often worsens both calibration and score alignment with the Beta prior, since the calibration map cannot approximate the identity function (as seen in datasets adult, bank, default, drybean, occupancy, and spambase). In this case, Beta calibration and isotonic regression frequently outperform Platt scaling in both calibration and alignment with the Beta prior. Notably, Beta calibration surpasses isotonic regression in most datasets, particularly concerning KL divergence. For models with initially uncalibrated scores (High ICI), post-calibration techniques either show lower ICI and lower KL divergence (abalone, coupon), or result in increased KL divergence alongside improved calibration (mushroom, occupancy). In such cases, all calibration methods exhibit similar trends in KL divergence and ICI, with no single post-calibration technique consistently outperforming the others, as their effectiveness varies across datasets.

To summarize, for already calibrated scores, post-calibration techniques generally reduce score alignment with Beta priors, as indicated by KL divergence, although Beta calibration results in a smaller deterioration compared to isotonic regression and Platt scaling. For scores with high initial ICI, post-calibration improves calibration but may either reduce or increase score alignment depending on the dataset.

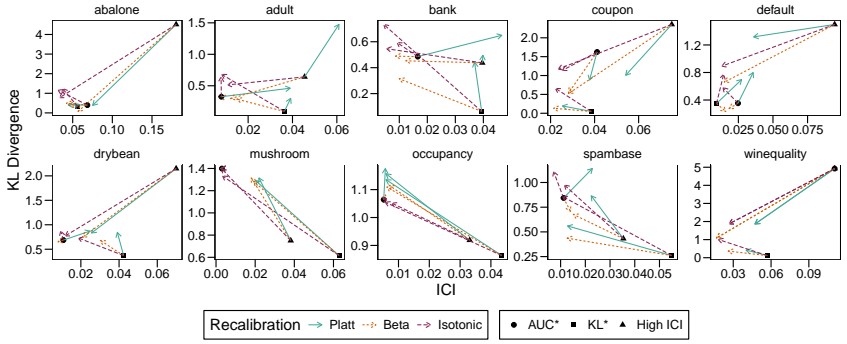

Notes: AUC*, KL*, High ICI: models selected by optimizing AUC, KL divergence, or by selecting a high ICI.

Figure 2: Average KL divergence and ICI before and after recalibration.

---

[1]For illustration purposes, the parameters of the prior distribution $\mathcal{B}$ are estimated via maximum likelihood using scores from a GAMSEL model [6], where the event is regressed on the variables generating the data.

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

# A  Platt scaling

## A.1  Reduction in Score Range

Platt scaling learns parameters, $\mu$ and $s$ ($s > 0$ for a non-decreasing calibration map g) on a calibration set. The obtained calibrated probabilities are:

$$\mathrm{g}\left(\hat{s}(\mathbf{x})\right) = \frac{1}{1 + \exp\left\{-\frac{1}{s}\left(\hat{s}(\mathbf{x}) - \mu\right)\right\}}.$$

With Platt scaling, the range of calibrated predicted scores is always narrower than the range of the initial scores when the parameter $s \geq \frac{1}{4}$. Indeed, since $\rho = \frac{1}{4}$ is the minimum value for which $\sigma(x) = \frac{1}{1 + \exp{-x}}$ remains $\rho$-Lipschitz on $\mathbb{R}$, for $x_1 < x_2 \in \mathbb{R}$, we have:

$$\left|\sigma\left(\frac{x_2 - \mu}{s}\right) - \sigma\left(\frac{x_1 - \mu}{s}\right)\right| \leq \frac{1}{4}\left|\frac{x_2 - \mu}{s} - \frac{x_1 - \mu}{s}\right| \leq \frac{1}{4s}\left|x_2 - x_1\right| \text{ with } s > 0.$$

As a result, if $s \geq \frac{1}{4}$, the range of $(\hat{s}(\mathbf{x}_i))_{i=1}^n$ is larger than the range of the calibrated scores with Platt scaling $(g(\hat{s}(\mathbf{x}_i)))_{i=1}^n$. Let $\hat{s}_m$ (resp. $\hat{s}_M$) denote the minimum (resp. the maximum) value of $(\hat{s}(\mathbf{x}_i))_{i=1}^n$. If $s \geq \frac{1}{4}$, we have:

$$\left| \frac{g(\hat{s}_M) - g(\hat{s}_m)}{\hat{s}_M - \hat{s}_m} \right| = \left| \frac{\sigma\left(\frac{\hat{s}_M - \mu}{s}\right) - \sigma\left(\frac{\hat{s}_m - \mu}{s}\right)}{\hat{s}_M - \hat{s}_m} \right| \leq 1.$$

And, due to the concavity of the sigmoid function over $[0, +\infty]$, this post-calibration method tends to reduce the range of predicted scores more significantly when the initial scores are highly concentrated.

# B Data

## B.1 Simulated Data

To simulate data, we consider the DGPs from Fernandes Machado et al. [9]. The first three are from Ojeda et al. [26]. In the fourth, an interaction term between two predictors is added. Each scenario uses a logistic model to generate the outcome. Let $Y_i$ be a binary variable following a Bernoulli distribution: $Y_i \sim B(p_i)$, where $p_i$ is the probability of observing $Y_i = 1$. The probability $p_i$ is defined by:

$$p_i = \mathbb{P}(Y = 1 \mid \mathbf{x}_i) = \left[1 + \exp(-\eta_i)\right]^{-1}. \tag{B.2}$$

For the second DGP, to introduce non-linearities, $p^3$ is used as true probabilities instead of $p$.

For all DGPs, $\eta_i = \mathbf{x}_i^\top \boldsymbol{\beta}$, where $\mathbf{x}_i$ is a vector of covariates and $\boldsymbol{\beta}$ is a vector of arbitrary scalars. The covariate vector includes two continuous predictors for DGPs 1 and 2. For DGP 3, it includes five continuous and five categorical predictors. For DGP 4, it contains three continuous variables, the square of the first variable, and an interaction term between the second and third variables. Specifically, $\eta_i = \beta_1 x_{1,i} + \beta_2 x_{2,i} + \beta_3 x_{3,i} + \beta_4 x_{1,i}^2 + \beta_5 x_{2,i} \times x_{3,i}$. Continuous predictors are drawn from $\mathcal{N}(0,1)$. Categorical predictors consist of two variables with two categories, one with three categories, and one with five categories, all uniformly distributed. The values of coefficients $\boldsymbol{\beta}$ are reported in Table B1.

For each DGP, we generate data considering four scenarios with varying numbers of noise variables: 0, 10, 50, or 100 variables drawn from $\mathcal{N}(0,1)$.

For the fourth DGP, to achieve a similar probability distribution to DGP 1, we perform resampling using a rejection algorithm (the algorithm is detailed in [9]).

| DGP | No. Cont. | No. Cat. | No. Noise | $\boldsymbol{\beta}$ | Type $\eta$ |
|---|---|---|---|---|---|
| 1 | 2 | 0 | $\{0, 10, 50, 100\}$ | $(.5, 1)$ | Linear terms |
| 2 | | | Same as DGP 2, but with probabilities $p^3$ | | |
| 3 | 5 | 5 | $\{0, 10, 50, 100\}$ | $(.1, .2, .3, .4, .5, .01, .02, .03, .04, .05)$ | Linear terms |
| 4 | 3 | 0 | $\{0, 10, 50, 100\}$ | $(.5, 1, .3)$ | Non-linear terms |

Notes: No. Cont., No. Cat. and No. Noise correspond to the number of continuous, categorical and noise variables, respectively.

Table B1: Parameters of the different scenarios.

The datasets are split into four parts: a training sample, a validation sample, a calibration sample, and a test sample, each containing 10,000 observations. The empirical distribution of samples of from each DGP are shown in Fig. B1.

## B.2 Real Data

The main characteristics of the datasets are summarized in Table B2.

Most of the datasets used are associated with classification tasks. If not, they contain a binary variable suitable for classification or a variable that can be converted into a binary variable. The target variables for each dataset are as follows:

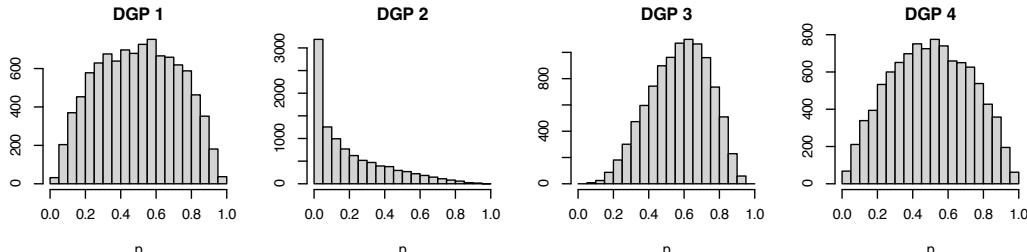

Figure B1: Distribution of the underlying probabilities in the different categories of scenarios.

Table B2: Key characteristics of the datasets

| Dataset | $n$ | No. predictors | No. num. predictors | Prop. target = 1 | Reference | License |
|---|---|---|---|---|---|---|
| abalone | 4,177 | 8 | 8 | 0.37 | Nash et al. [25] | CC BY 4.0 |
| adult | 32,561 | 14 | 6 | 0.24 | Becker and Kohavi [2] | CC BY 4.0 |
| bank | 45,211 | 16 | 7 | 0.12 | Moro et al. [23] | CC BY 4.0 |
| default | 30,000 | 23 | 14 | 0.22 | Yeh [36] | CC BY 4.0 |
| drybean | 13,611 | 16 | 16 | 0.26 | Koklu and Ali Ozkan [15] | CC BY 4.0 |
| coupon | 12,079 | 22 | 0 | 0.57 | Wang et al. [34] | CC BY 4.0 |
| mushroom | 8,124 | 21 | 0 | 0.52 | Schlimmer [32] | CC BY 4.0 |
| occupancy | 20,560 | 5 | 5 | 0.23 | Candanedo [5] | CC BY 4.0 |
| winequality | 6,495 | 12 | 11 | 0.63 | Cortez et al. [7] | CC BY 4.0 |
| spambase | 4,601 | 57 | 57 | 0.39 | Hopkins et al. [14] | CC BY 4.0 |

Notes: $n$ represents the number of observations, 'No. predictors' the total number of predictors, 'No. num. predictors' the number of numeric predictors, and 'Prop. target = 1' the proportion of positive observed events.

- `abalone`: gender of abalones (1 for male, 0 for female); originally used to predict the size of abalones.

- `adult`: high income (1 if income $\geq$ 50k per year).

- `bank`: subscription to a term deposit (1 if yes, 0 otherwise).

- `default`: default payment (1 if default, 0 otherwise).

- `drybean`: type of dry bean (1 if dermason, 0 otherwise); originally a multi-class variable.

- `coupon`: acceptance of a recommended coupon in different driving scenarios (1 if accepted, 0 otherwise).

- `mushroom`: mushroom classification (1 if edible, 0 otherwise).

- `occupancy`: prediction of room occupancy (1 if occupied, 0 otherwise); originally aimed at predicting the age of occupancy from physical measurements.

- `winequality`: quality of wine (1 if quality $\geq$ 6, 0 otherwise); originally a scale from 0 to 10, with 0 being bad quality and 10 being good quality.

- `spambase`: email classification (1 if spam, 0 otherwise).

## C  Numerical Experiments

### C.1  Simulated Data

For each of the four DGPs (see Section B.1) and each configuration of the number of noise variables (0, 10, 50, or 100), we generate 100 sample replications. For each sample, we train an XGBoost model on 10,000 observation using the `xgb.train` function from the R package `xgboost`. The learning rate is set to 0.3. The tree depth (argument `max_depth`) varies according to the following values: 2, 4, 6. The number of boosting iterations (argument `nrounds`) ranges from 1 to 400. All variables (predictors and, if applicable, noise variables) are included in the model without transformation.

For each model configuration, we select the hyperparameters based on different criteria using the validation set results. Specifically, we make three model choices:

- AUC*: hyperparameters are selected to maximize the AUC.

- KL*: hyperparameters are chosen to minimize the Kullback-Leibler divergence between the scores on the validation set and the true probability distribution (observable here in the context of simulated data).

- High ICI: hyperparameters are selected to produce relatively poor calibration, as measured by the ICI. Specifically, we select the model with the smallest ICI among those with an ICI at least one standard deviation above the mean ICI obtained during grid search.

Once the hyperparameters are selected, we apply a recalibration method on an independent calibration set: either Platt scaling, Beta calibration, or isotonic regression.

The model performance is then evaluated on a test set, allowing for comparison based on: (i) the metric used to select the hyperparameters, and (ii) whether or not calibration techniques were applied to the scores.

Figs C1 to C16 display the empirical distribution of scores for a single replication (the first one) in each of the $4 \times 4$ configurations (4 DGPs and 4 different values for the number of noise variables introduced in the training data). In each figure, the first row shows the distribution of test set scores without applying any calibration technique to the selected model. The second row, in green, shows the score distributions after applying Platt scaling for calibration. The third row, in orange, shows the score distribution after applying Beta calibration, and, lastly, the fourth row, in purple, displays the score distributions after applying isotonic regression. The columns correspond to the criteria used to select the hyperparameters based on the validation set results: AUC, Brier score, ICI, KL, or a set chosen such that the ICI is high.

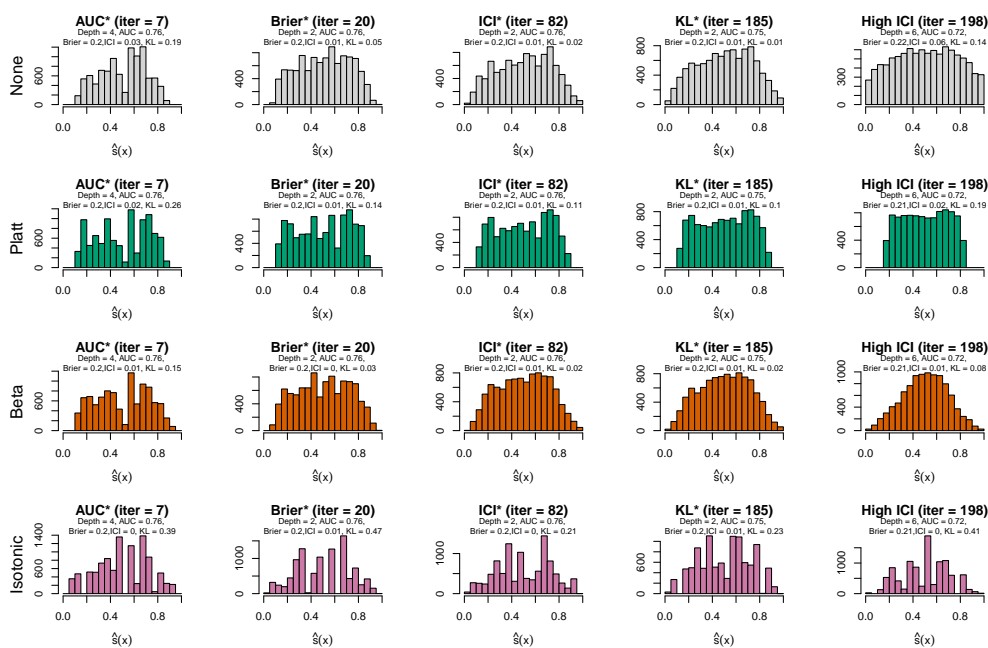

Figure C1: Distribution of estimated scores for XGB: **DGP 1**, **0 noise variable**, single replication.

Notes: AUC*, Brier*, ICI*, and KL*: models selected based on optimizing AUC, Brier score, ICI, and Kullback-Leibler divergence, resp.

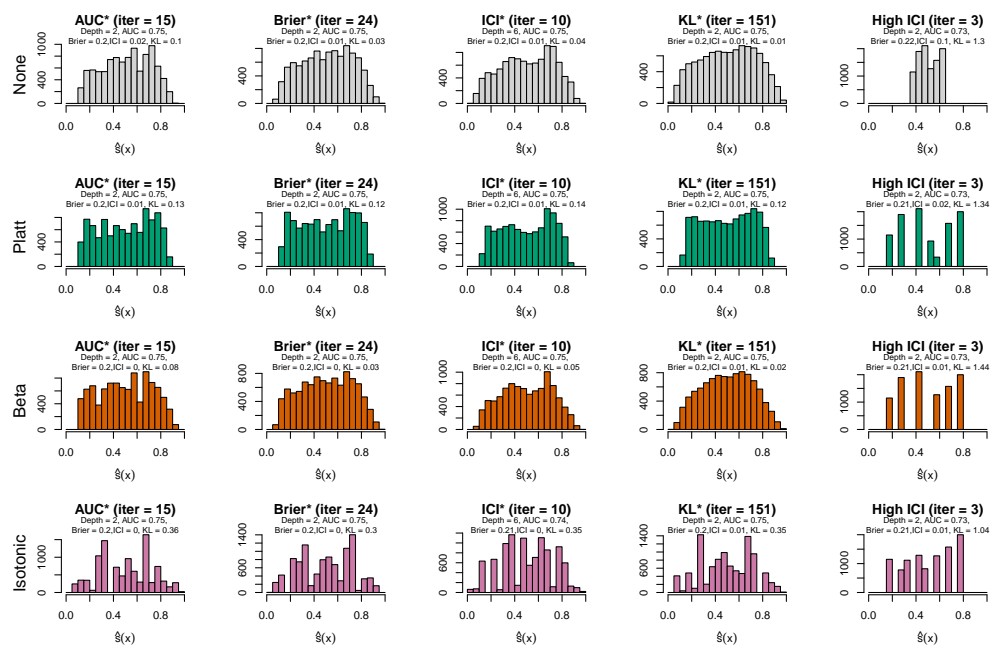

Figure C2: Distribution of estimated scores for XGB: **DGP 1**, **10 noise variables**, single replication.

Notes: AUC*, Brier*, ICI*, and KL*: models selected based on optimizing AUC, Brier score, ICI, and Kullback-Leibler divergence, resp.

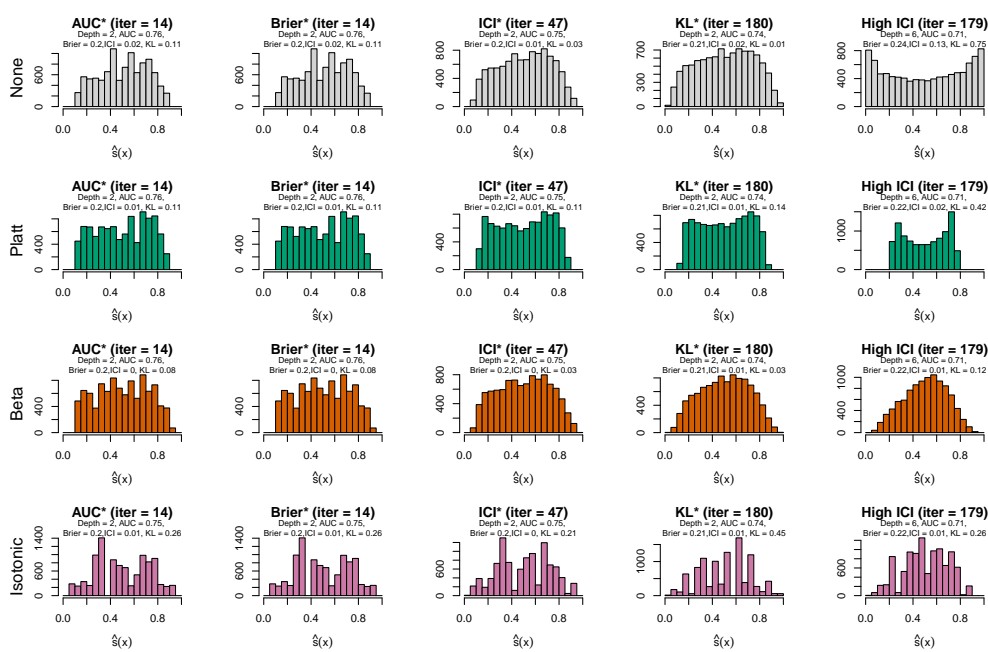

Figure C3: Distribution of estimated scores for XGB: **DGP 1**, **50 noise variables**, single replication.

Notes: AUC*, Brier*, ICI*, and KL*: models selected based on optimizing AUC, Brier score, ICI, and Kullback-Leibler divergence, resp.

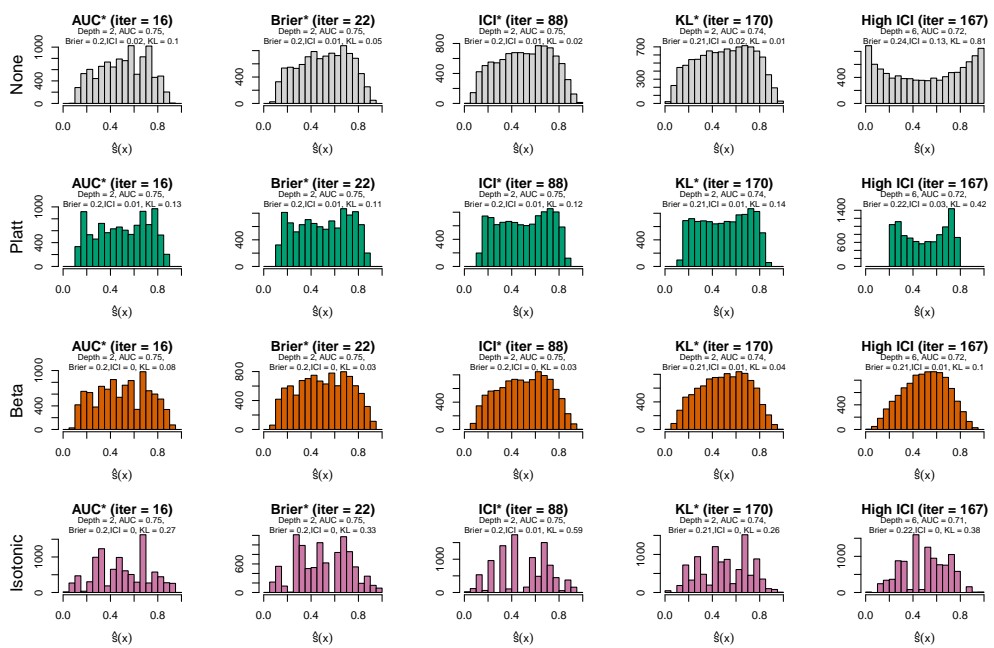

Figure C4: Distribution of estimated scores for XGB: **DGP 1**, **100 noise variables**, single replication.

Notes: AUC*, Brier*, ICI*, and KL*: models selected based on optimizing AUC, Brier score, ICI, and Kullback-Leibler divergence, resp.

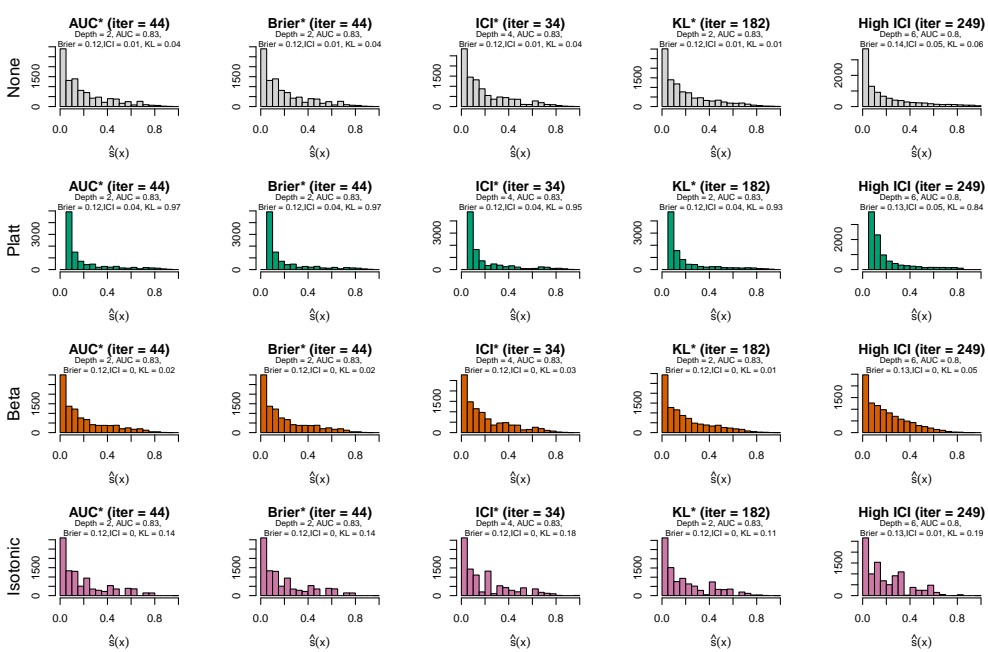

Figure C5: Distribution of estimated scores for XGB: **DGP 2**, **0 noise variable**, single replication.

Notes: AUC*, Brier*, ICI*, and KL*: models selected based on optimizing AUC, Brier score, ICI, and Kullback-Leibler divergence, resp.

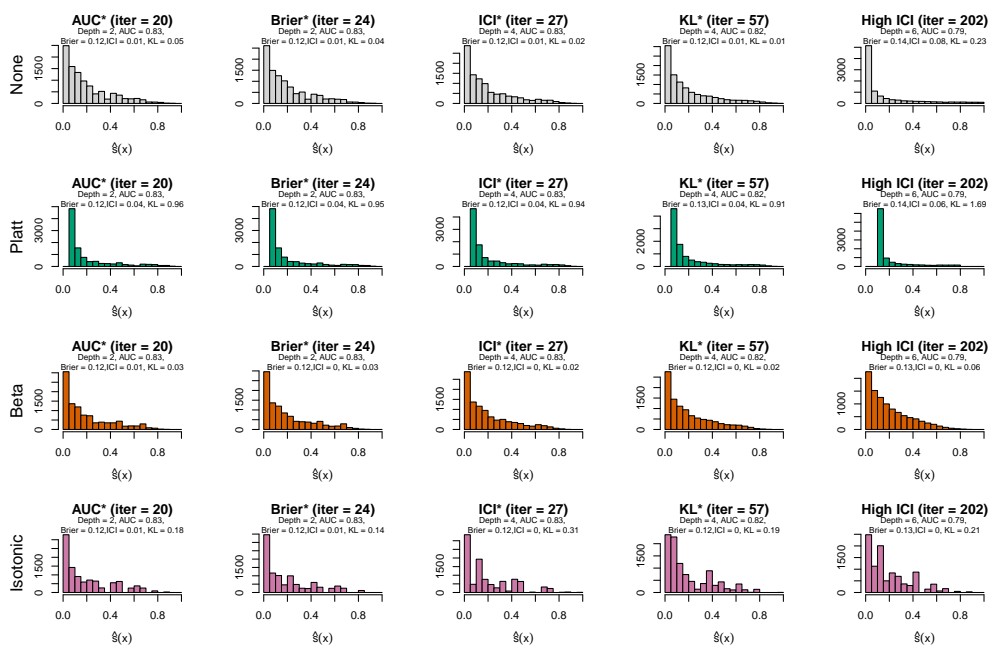

Figure C6: Distribution of estimated scores for XGB: **DGP 2**, **10 noise variables**, single replication.

Notes: AUC*, Brier*, ICI*, and KL*: models selected based on optimizing AUC, Brier score, ICI, and Kullback-Leibler divergence, resp.

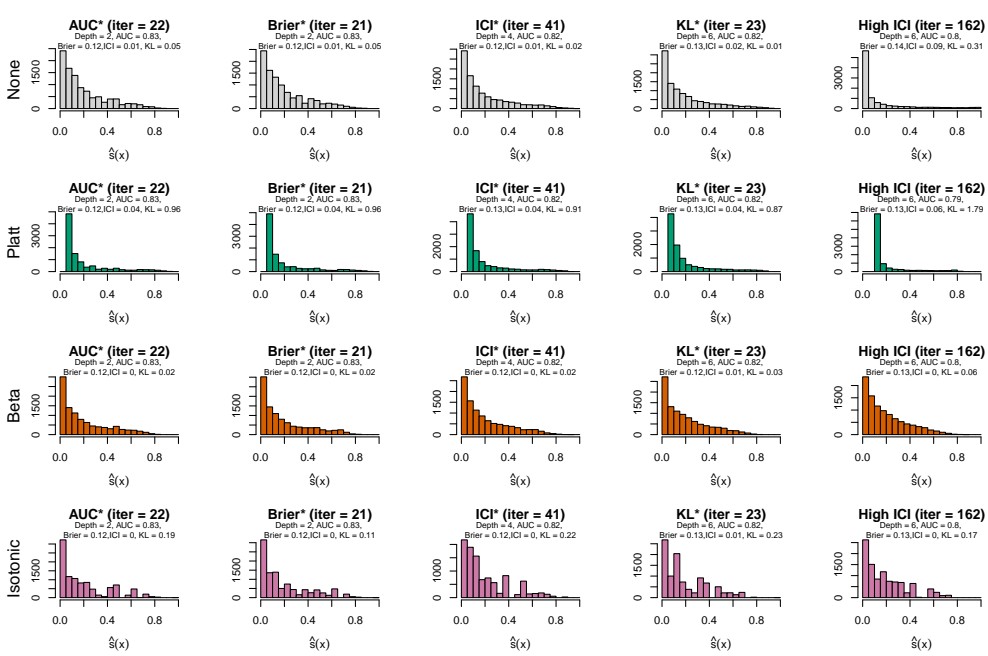

Figure C7: Distribution of estimated scores for XGB: **DGP 2**, **50 noise variables**, single replication.

Notes: AUC*, Brier*, ICI*, and KL*: models selected based on optimizing AUC, Brier score, ICI, and Kullback-Leibler divergence, resp.

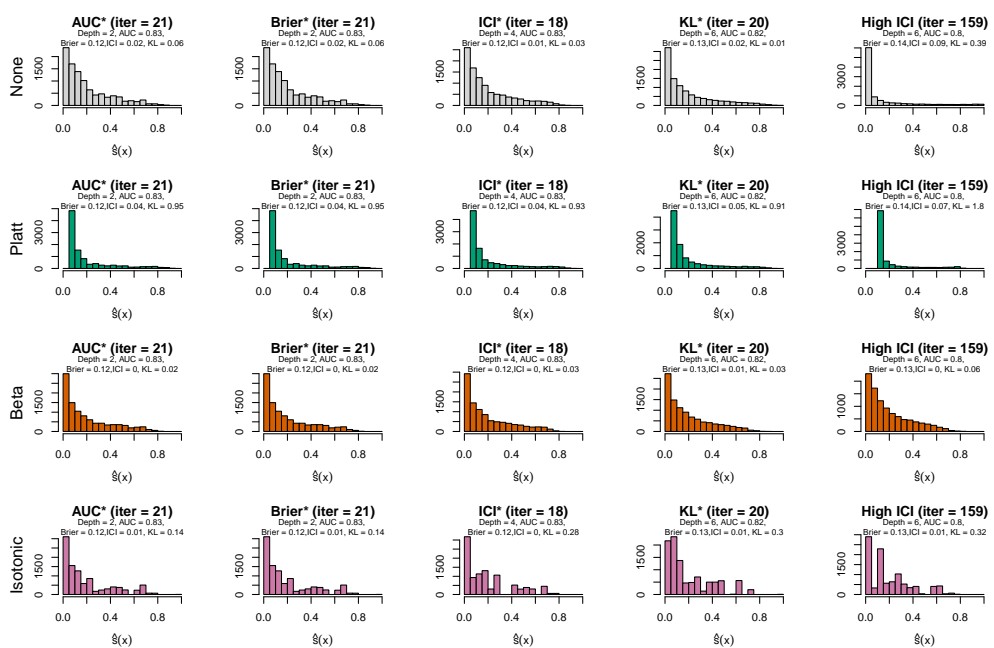

Figure C8: Distribution of estimated scores for XGB: **DGP 2**, **100 noise variables**, single replication.

Notes: AUC*, Brier*, ICI*, and KL*: models selected based on optimizing AUC, Brier score, ICI, and Kullback-Leibler divergence, resp.

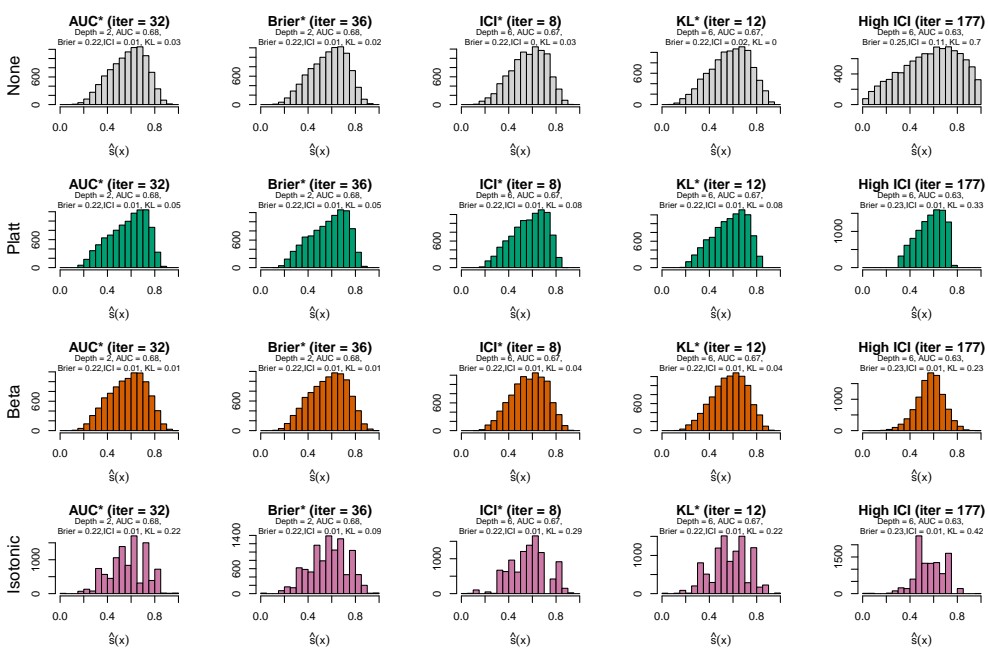

Figure C9: Distribution of estimated scores for XGB: **DGP 3**, **0 noise variable**, single replication.

Notes: AUC*, Brier*, ICI*, and KL*: models selected based on optimizing AUC, Brier score, ICI, and Kullback-Leibler divergence, resp.

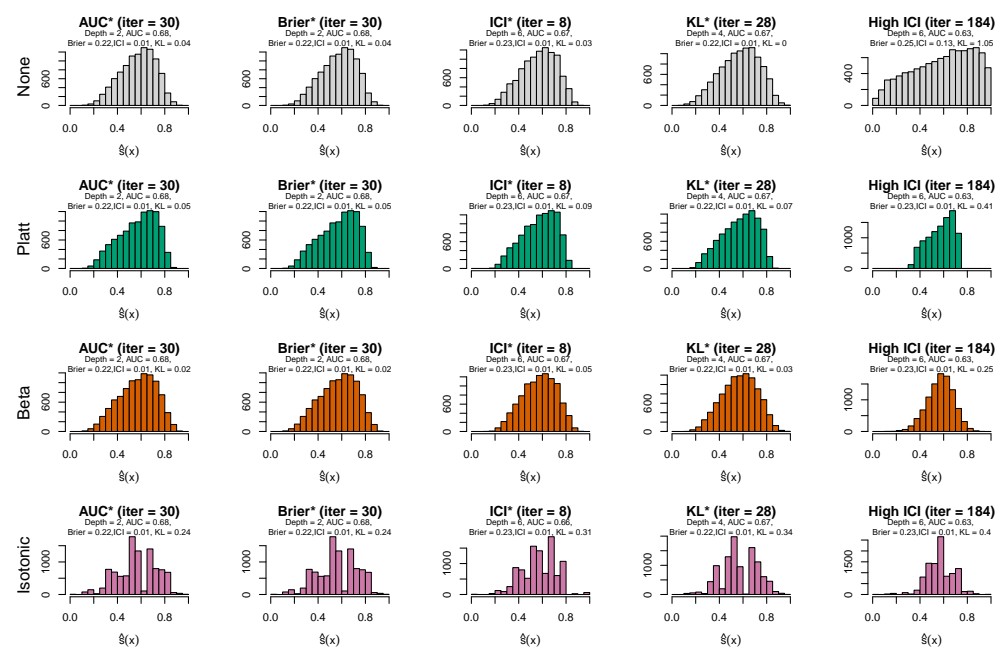

Figure C10: Distribution of estimated scores for XGB: **DGP 3**, **10 noise variables**, single replication.

Notes: AUC*, Brier*, ICI*, and KL*: models selected based on optimizing AUC, Brier score, ICI, and Kullback-Leibler divergence, resp.

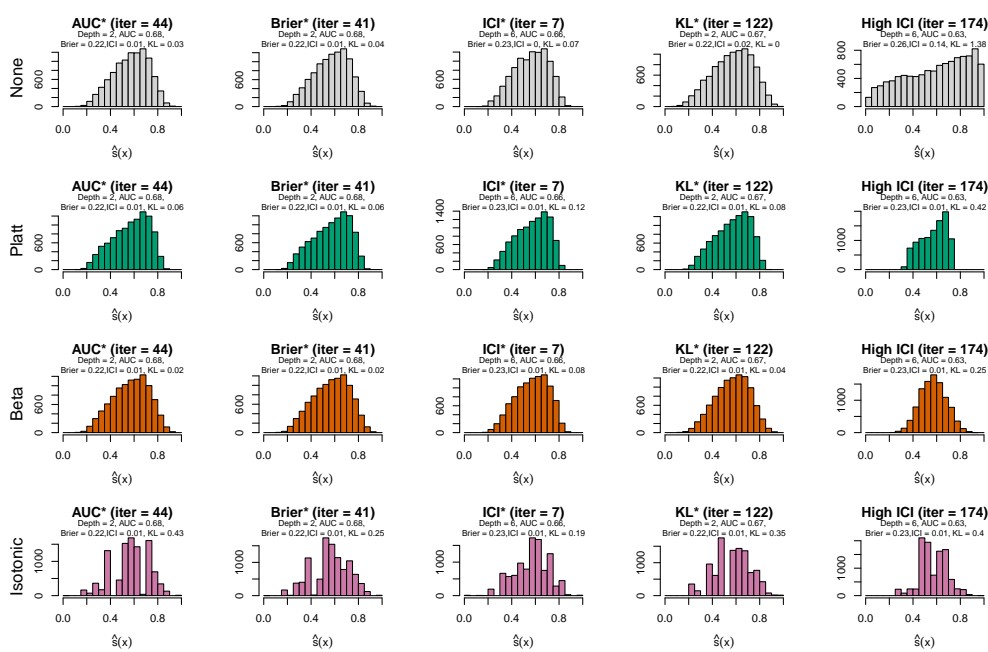

Figure C11: Distribution of estimated scores for XGB: **DGP 3**, **50 noise variables**, single replication.

Notes: AUC*, Brier*, ICI*, and KL*: models selected based on optimizing AUC, Brier score, ICI, and Kullback-Leibler divergence, resp.

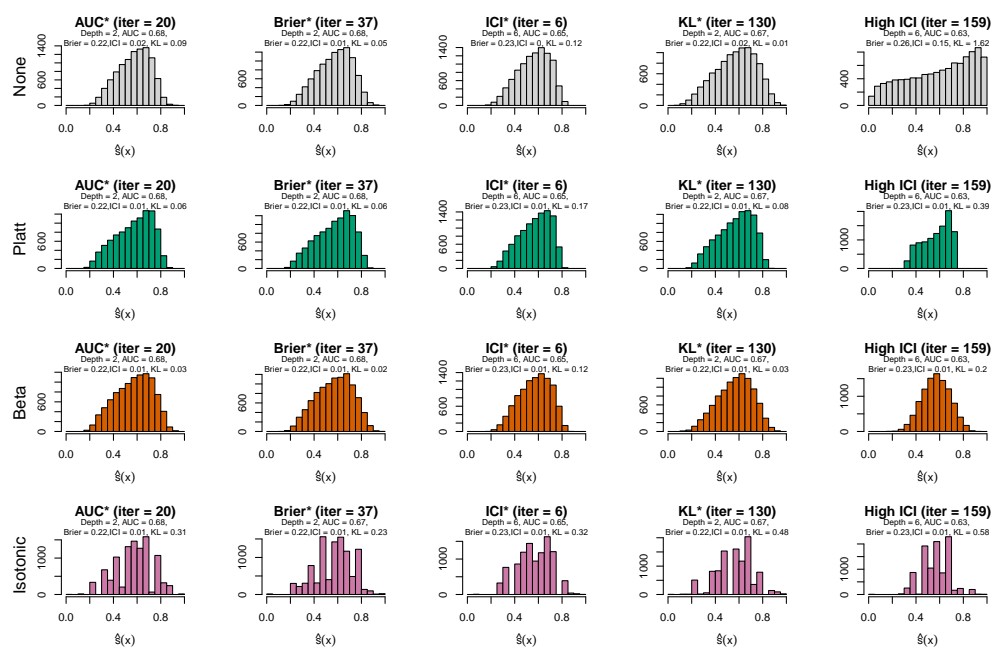

Figure C12: Distribution of estimated scores for XGB: **DGP 3**, **100 noise variables**, single replication.

Notes: AUC*, Brier*, ICI*, and KL*: models selected based on optimizing AUC, Brier score, ICI, and Kullback-Leibler divergence, resp.

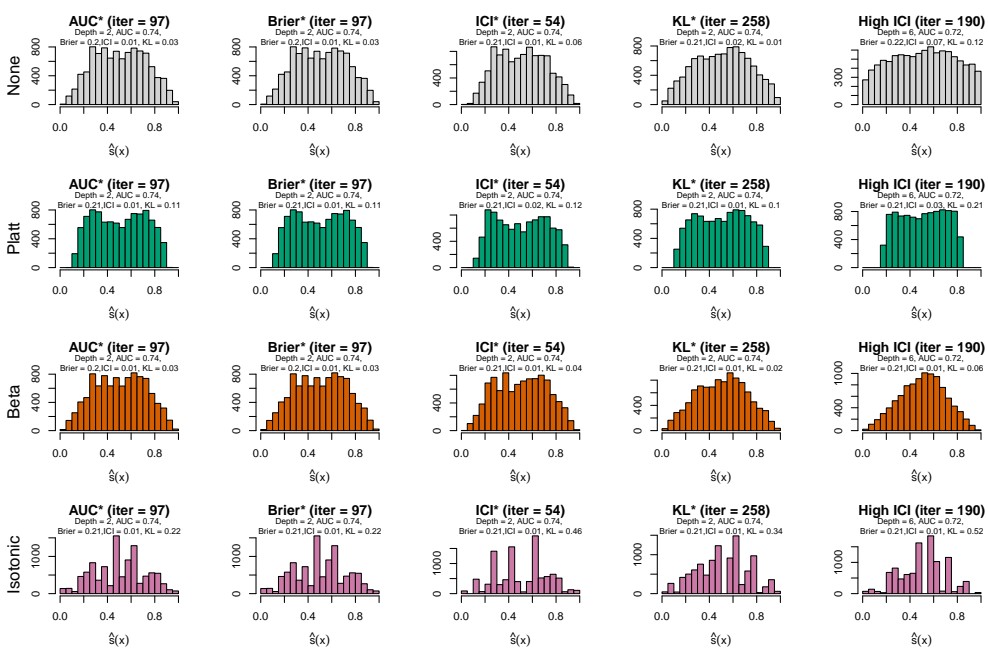

Figure C13: Distribution of estimated scores for XGB: **DGP 4**, **0 noise variable**, single replication.

Notes: AUC*, Brier*, ICI*, and KL*: models selected based on optimizing AUC, Brier score, ICI, and Kullback-Leibler divergence, resp.

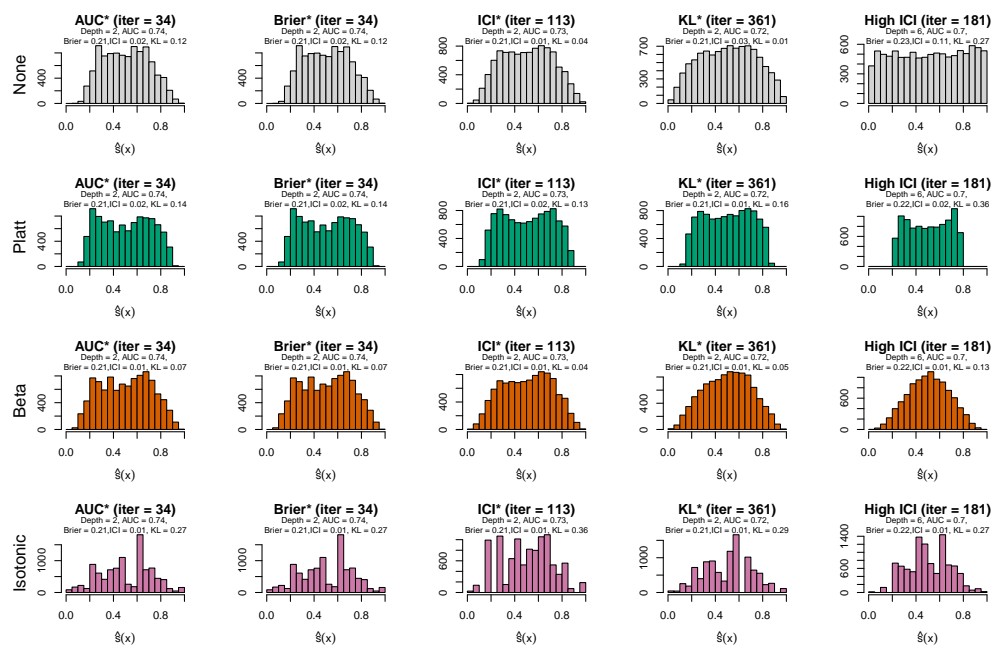

Figure C14: Distribution of estimated scores for XGB: **DGP 4**, **10 noise variables**, single replication.

Notes: AUC*, Brier*, ICI*, and KL*: models selected based on optimizing AUC, Brier score, ICI, and Kullback-Leibler divergence, resp.

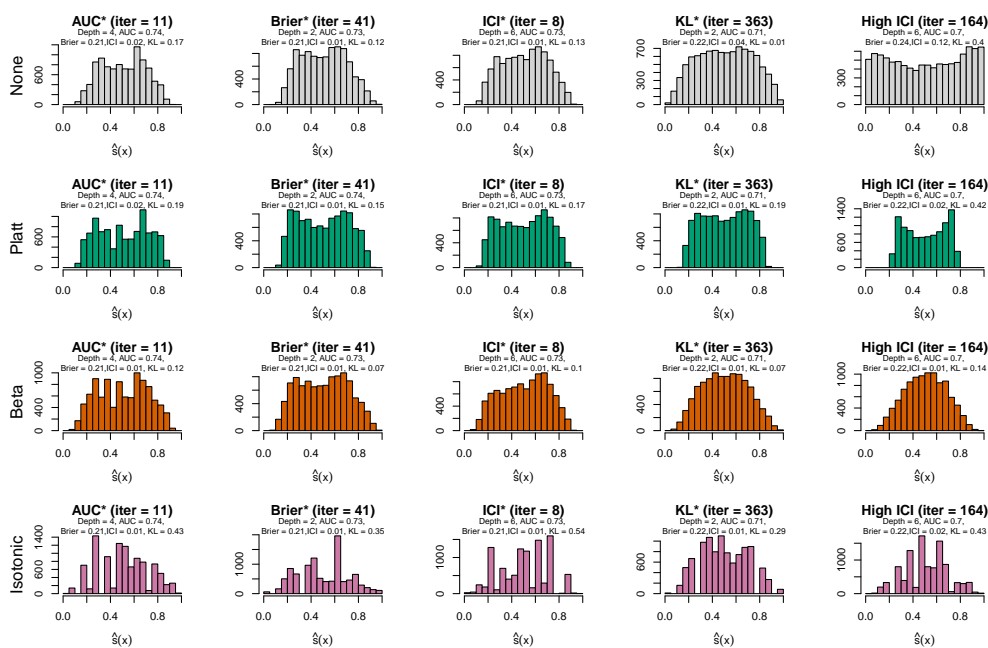

Figure C15: Distribution of estimated scores for XGB: **DGP 4**, **50 noise variables**, single replication.

Notes: AUC*, Brier*, ICI*, and KL*: models selected based on optimizing AUC, Brier score, ICI, and Kullback-Leibler divergence, resp.

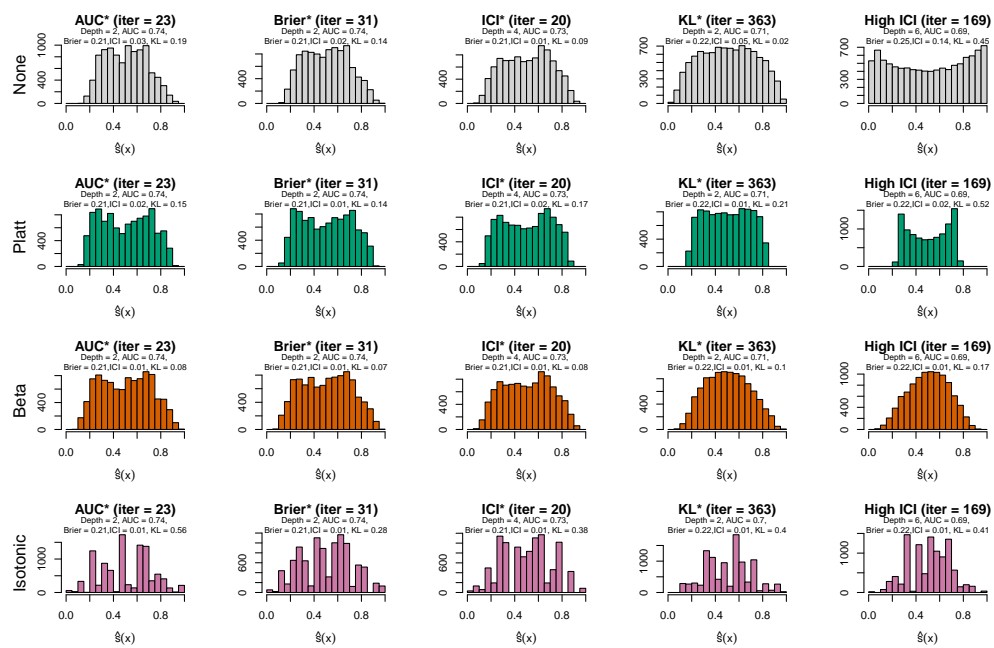

Figure C16: Distribution of estimated scores for XGB: **DGP 4**, **100 noise variables**, single replication.

Notes: AUC*, Brier*, ICI*, and KL*: models selected based on optimizing AUC, Brier score, ICI, and Kullback-Leibler divergence, resp.

Table C1 reports the average values of metrics calculated on the test set over 100 replications for each DGP and each number of noise variables in the training data. The values are presented for models selected by optimizing, on the validation set, either AUC (AUC*) or KL divergence (KL*), as well as for a model with poor calibration (High ICI). The metrics are calculated before applying any calibration method (column "None"), after applying Platt scaling, Beta calibration and isotonic regression calibration.

Fig. C17 shows the performance of the models, measured by the KL divergence between the test set score distribution (x-axis) and the true probability distribution (y-axis), before and after applying calibration methods. The values represent the average of these two metrics over 100 replications for each DGP (rows), based on the number of noise variables in the training set (columns). The point corresponds to the model whose hyperparameters (number of boosting iterations and tree depth) are selected to maximize AUC on the validation set. The square represents the model selected by minimizing the Kullback-Leibler divergence between the score distribution on the validation set and the true probability distribution. The triangle denotes a model with poor calibration on the test set. Solid green arrows illustrate the change in metrics after applying Platt scaling calibration, dotted orange arrows show changes after applying Beta calibration, and dashed purple arrows indicate changes after applying isotonic regression calibration.

## C.2 Real Data

We train XGBoost models on the 10 datasets presented in Section B.2. Unlike Section C.1, the true probabilities underlying the binary events are not observable. Here, we assume that we have prior knowledge about the probability distribution, which can be considered as expert opinion. To simulate this expert opinion, we assume that the true probabilities follow a Beta distribution. The parameters of this distribution, specific to each dataset, are estimated via MLE using the scores from a GAMSEL model [6].

Using these prior distributions, it is possible to replicate the estimation procedure previously applied to the simulated data. Each dataset is split into two parts: 80% of the observations are used to train an XGBoost model (on a training set comprising 70% of these observations, with hyperparameters selected based on metrics calculated on the remaining 20% validation set), and the remaining 30%

Table C1: Comparison of metrics computed on the validation set for models selected based on AUC, KL divergence, or ICI across 100 replications. Standard errors are provided in parentheses.

| | | | None | | | Platt scaling | | | Beta | | | Isotonic | | |
|---|---|---|---|---|---|---|---|---|---|---|---|---|---|---|
| DGP | Noise | Optim. | BS | ICI | KL | BS | ICI | KL | BS | ICI | KL | BS | ICI | KL |
| 1 | 0 | AUC* | .201 (.002) | .011 (.005) | .051 (.03) | .201 (.002) | .017 (.005) | .131 (.031) | .201 (.002) | .011 (.004) | .051 (.024) | .201 (.002) | .012 (.004) | .304 (.095) |
| | | KL* | .202 (.002) | .013 (.005) | .021 (.006) | .202 (.002) | .015 (.005) | .107 (.016) | .202 (.002) | .011 (.004) | .027 (.011) | .202 (.002) | .012 (.004) | .307 (.101) |
| | | High ICI | .217 (.002) | .062 (.006) | .158 (.066) | .213 (.002) | .018 (.005) | .199 (.05) | .212 (.002) | .013 (.004) | .08 (.053) | .212 (.002) | .011 (.004) | .348 (.104) |
| | 10 | AUC* | .201 (.002) | .014 (.005) | .063 (.032) | .201 (.002) | .018 (.005) | .135 (.025) | .201 (.002) | .011 (.004) | .053 (.024) | .201 (.002) | .011 (.004) | .296 (.101) |
| | | KL* | .204 (.002) | .015 (.005) | .01 (.004) | .204 (.002) | .016 (.005) | .109 (.015) | .203 (.002) | .01 (.004) | .017 (.008) | .204 (.002) | .012 (.004) | .302 (.11) |
| | | High ICI | .229 (.003) | .106 (.006) | .442 (.194) | .216 (.002) | .025 (.005) | .386 (.102) | .215 (.002) | .011 (.004) | .106 (.144) | .215 (.002) | .012 (.004) | .364 (.137) |
| | 50 | AUC* | .201 (.002) | .016 (.005) | .08 (.032) | .201 (.002) | .018 (.005) | .142 (.029) | .201 (.002) | .012 (.004) | .06 (.027) | .201 (.002) | .012 (.004) | .304 (.098) |
| | | KL* | .205 (.002) | .019 (.005) | .009 (.003) | .205 (.002) | .016 (.004) | .12 (.02) | .205 (.002) | .01 (.004) | .022 (.01) | .205 (.002) | .012 (.004) | .313 (.095) |
| | | High ICI | .235 (.003) | .129 (.006) | .717 (.169) | .216 (.002) | .029 (.006) | .453 (.166) | .215 (.002) | .01 (.004) | .117 (.223) | .215 (.002) | .011 (.004) | .359 (.211) |
| | 100 | AUC* | .201 (.002) | .016 (.005) | .087 (.029) | .201 (.002) | .018 (.005) | .144 (.024) | .201 (.002) | .012 (.004) | .061 (.024) | .201 (.002) | .011 (.004) | .324 (.114) |
| | | KL* | .206 (.002) | .019 (.005) | .009 (.004) | .206 (.002) | .015 (.004) | .125 (.023) | .206 (.002) | .01 (.004) | .025 (.012) | .206 (.002) | .011 (.004) | .302 (.101) |
| | | High ICI | .236 (.003) | .136 (.006) | .807 (.093) | .216 (.002) | .031 (.005) | .444 (.032) | .215 (.002) | .01 (.004) | .086 (.055) | .215 (.002) | .011 (.004) | .343 (.115) |
| 2 | 0 | AUC* | .118 (.002) | .01 (.004) | .029 (.014) | .12 (.002) | .038 (.004) | .783 (.217) | .118 (.002) | .009 (.004) | .027 (.012) | .118 (.002) | .009 (.004) | .214 (.069) |
| | | KL* | .12 (.002) | .01 (.004) | .013 (.005) | .121 (.002) | .038 (.004) | .863 (.141) | .119 (.002) | .009 (.004) | .016 (.007) | .12 (.002) | .009 (.004) | .215 (.073) |
| | | High ICI | .131 (.003) | .048 (.004) | .128 (.125) | .13 (.003) | .05 (.005) | .845 (.042) | .127 (.003) | .009 (.004) | .046 (.014) | .127 (.003) | .009 (.003) | .237 (.073) |
| | 10 | AUC* | .119 (.002) | .012 (.004) | .03 (.016) | .12 (.002) | .038 (.004) | .759 (.221) | .118 (.002) | .01 (.003) | .026 (.011) | .119 (.002) | .009 (.004) | .213 (.074) |
| | | KL* | .12 (.002) | .011 (.003) | .007 (.003) | .122 (.002) | .04 (.004) | .887 (.076) | .12 (.002) | .009 (.004) | .012 (.006) | .121 (.002) | .01 (.004) | .205 (.074) |
| | | High ICI | .137 (.003) | .075 (.004) | .288 (.127) | .132 (.002) | .058 (.004) | 1.24 (.354) | .128 (.002) | .009 (.004) | .045 (.029) | .128 (.002) | .009 (.003) | .263 (.1) |
| | 50 | AUC* | .119 (.002) | .013 (.004) | .038 (.018) | .12 (.002) | .038 (.004) | .728 (.221) | .119 (.002) | .01 (.004) | .029 (.013) | .119 (.002) | .009 (.004) | .207 (.069) |
| | | KL* | .121 (.002) | .011 (.003) | .006 (.003) | .123 (.002) | .041 (.004) | .894 (.045) | .121 (.002) | .009 (.003) | .014 (.008) | .121 (.002) | .009 (.004) | .215 (.068) |
| | | High ICI | .139 (.003) | .089 (.004) | .429 (.105) | .133 (.003) | .064 (.005) | 1.799 (.155) | .127 (.002) | .01 (.004) | .041 (.02) | .127 (.002) | .009 (.003) | .235 (.073) |
| | 100 | AUC* | .119 (.002) | .014 (.004) | .044 (.023) | .121 (.002) | .038 (.004) | .729 (.224) | .119 (.002) | .01 (.004) | .03 (.013) | .119 (.002) | .009 (.003) | .214 (.062) |
| | | KL* | .122 (.002) | .012 (.004) | .006 (.003) | .124 (.002) | .041 (.004) | .89 (.033) | .122 (.002) | .009 (.004) | .016 (.008) | .122 (.002) | .009 (.003) | .217 (.076) |
| | | High ICI | .14 (.003) | .093 (.004) | .482 (.099) | .133 (.003) | .065 (.005) | 1.842 (.155) | .127 (.002) | .01 (.004) | .042 (.014) | .127 (.002) | .009 (.004) | .23 (.07) |
| 3 | 0 | AUC* | .22 (.002) | .01 (.004) | .012 (.009) | .221 (.002) | .012 (.004) | .041 (.013) | .22 (.002) | .01 (.004) | .013 (.008) | .221 (.002) | .011 (.004) | .268 (.106) |
| | | KL* | .221 (.002) | .012 (.004) | .005 (.002) | .221 (.002) | .011 (.004) | .047 (.014) | .221 (.002) | .01 (.004) | .017 (.011) | .222 (.002) | .011 (.004) | .286 (.115) |
| | | High ICI | .246 (.002) | .105 (.005) | .631 (.086) | .231 (.001) | .014 (.004) | .268 (.047) | .231 (.001) | .011 (.004) | .174 (.035) | .231 (.001) | .011 (.004) | .376 (.108) |
| | 10 | AUC* | .221 (.001) | .011 (.004) | .027 (.018) | .221 (.002) | .012 (.005) | .046 (.014) | .221 (.001) | .01 (.004) | .017 (.009) | .221 (.002) | .011 (.004) | .28 (.099) |
| | | KL* | .222 (.002) | .014 (.005) | .004 (.002) | .222 (.002) | .012 (.004) | .056 (.019) | .222 (.002) | .01 (.004) | .023 (.016) | .222 (.002) | .011 (.004) | .284 (.103) |
| | | High ICI | .253 (.003) | .127 (.005) | .932 (.118) | .232 (.001) | .015 (.004) | .366 (.035) | .232 (.001) | .01 (.004) | .191 (.04) | .232 (.001) | .011 (.004) | .392 (.108) |
| | 50 | AUC* | .221 (.001) | .013 (.005) | .053 (.03) | .221 (.002) | .012 (.004) | .049 (.015) | .221 (.002) | .01 (.004) | .021 (.011) | .222 (.002) | .011 (.004) | .29 (.124) |
| | | KL* | .224 (.002) | .018 (.006) | .004 (.002) | .224 (.002) | .011 (.004) | .075 (.023) | .224 (.002) | .01 (.004) | .037 (.021) | .224 (.002) | .011 (.004) | .284 (.104) |
| | | High ICI | .259 (.003) | .145 (.006) | 1.285 (.169) | .233 (.001) | .017 (.005) | .402 (.027) | .232 (.001) | .01 (.004) | .204 (.044) | .232 (.001) | .011 (.004) | .424 (.127) |
| | 100 | AUC* | .222 (.001) | .015 (.006) | .067 (.031) | .222 (.002) | .012 (.004) | .052 (.016) | .221 (.002) | .01 (.004) | .024 (.012) | .222 (.002) | .011 (.004) | .286 (.107) |
| | | KL* | .225 (.002) | .019 (.005) | .004 (.002) | .224 (.002) | .011 (.004) | .08 (.021) | .224 (.002) | .01 (.004) | .042 (.02) | .225 (.002) | .011 (.004) | .301 (.122) |
| | | High ICI | .261 (.003) | .152 (.005) | 1.454 (.18) | .233 (.001) | .017 (.004) | .418 (.036) | .232 (.001) | .01 (.004) | .206 (.038) | .233 (.001) | .011 (.004) | .416 (.11) |
| 4 | 0 | AUC* | .204 (.002) | .011 (.004) | .039 (.021) | .205 (.002) | .016 (.004) | .13 (.02) | .204 (.002) | .011 (.004) | .035 (.014) | .205 (.002) | .011 (.005) | .294 (.1) |
| | | KL* | .206 (.002) | .018 (.005) | .011 (.004) | .206 (.002) | .015 (.004) | .115 (.012) | .206 (.002) | .01 (.004) | .019 (.007) | .206 (.002) | .011 (.004) | .289 (.105) |
| | | High ICI | .222 (.003) | .073 (.006) | .199 (.286) | .215 (.003) | .019 (.006) | .249 (.191) | .215 (.003) | .011 (.004) | .113 (.215) | .215 (.003) | .012 (.005) | .36 (.222) |
| | 10 | AUC* | .206 (.002) | .014 (.005) | .089 (.026) | .206 (.002) | .016 (.005) | .142 (.022) | .206 (.002) | .012 (.004) | .06 (.017) | .206 (.002) | .012 (.005) | .294 (.104) |
| | | KL* | .211 (.002) | .028 (.005) | .014 (.005) | .21 (.002) | .015 (.004) | .156 (.02) | .21 (.002) | .011 (.004) | .043 (.012) | .21 (.002) | .011 (.005) | .307 (.091) |
| | | High ICI | .232 (.002) | .105 (.005) | .307 (.236) | .219 (.002) | .021 (.005) | .391 (.109) | .219 (.002) | .01 (.004) | .14 (.144) | .219 (.002) | .011 (.005) | .422 (.181) |
| | 50 | AUC* | .207 (.002) | .019 (.005) | .126 (.031) | .207 (.002) | .016 (.005) | .145 (.025) | .207 (.002) | .012 (.004) | .072 (.02) | .207 (.002) | .012 (.005) | .3 (.104) |
| | | KL* | .215 (.002) | .034 (.005) | .017 (.004) | .213 (.002) | .014 (.004) | .19 (.021) | .213 (.002) | .011 (.004) | .072 (.018) | .214 (.002) | .012 (.004) | .345 (.101) |
| | | High ICI | .238 (.003) | .125 (.006) | .422 (.047) | .221 (.002) | .024 (.005) | .424 (.039) | .22 (.002) | .011 (.005) | .134 (.023) | .22 (.002) | .012 (.005) | .401 (.119) |
| | 100 | AUC* | .208 (.002) | .021 (.007) | .145 (.035) | .207 (.002) | .015 (.005) | .147 (.025) | .207 (.002) | .012 (.004) | .078 (.021) | .207 (.002) | .012 (.005) | .334 (.109) |
| | | KL* | .216 (.002) | .037 (.006) | .017 (.004) | .215 (.002) | .013 (.005) | .202 (.022) | .215 (.002) | .011 (.004) | .084 (.021) | .215 (.002) | .012 (.005) | .367 (.107) |
| | | High ICI | .241 (.003) | .133 (.006) | .486 (.046) | .221 (.002) | .025 (.004) | .468 (.057) | .22 (.002) | .011 (.005) | .141 (.023) | .22 (.002) | .011 (.005) | .398 (.101) |

Notes: AUC*, KL*, High ICI: models selected by optimizing AUC, KL divergence, or by selecting a high ICI.

are used for model calibration (with 60% of this subset forming the calibration set) and for testing model performance on unseen data (the remaining 40%).

Table C2 presents the metrics calculated on the test set for models selected based on their validation performance, according to AUC (AUC*), KL divergence between the score distribution and the prior distribution (KL*), or to intentionally obtain poor calibration (High ICI), both before and after applying calibration methods. For real datasets, High ICI refers to the model with hyperparameters yielding the highest AUC among models with an ICI at least one standard deviation above the mean ICI observed during grid search. This table complements Fig. 2 from the main part of the article.

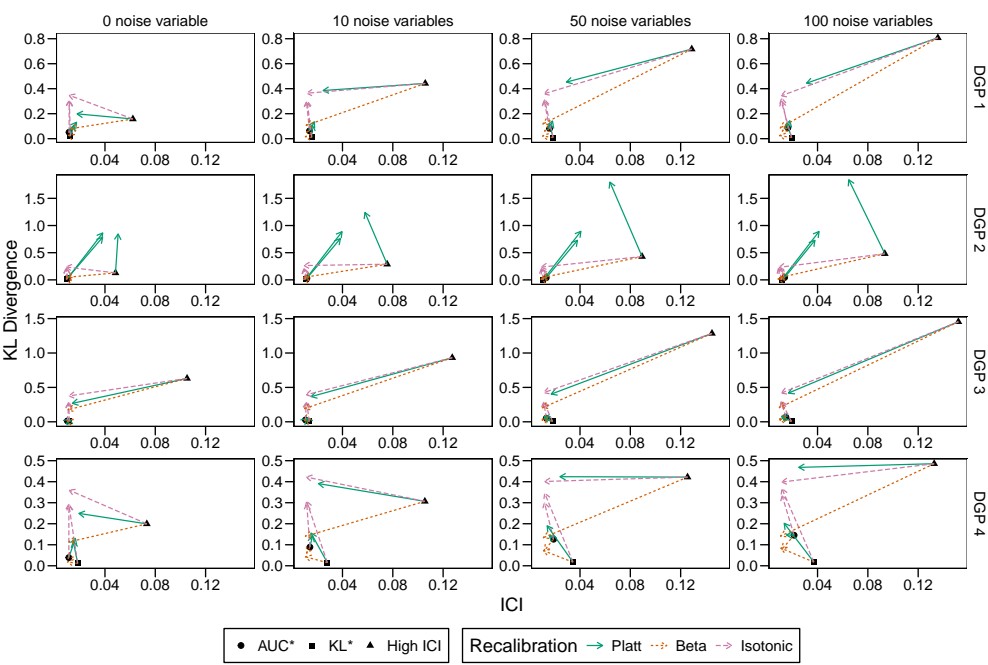

Figure C17: Average KL divergence and ICI before and after recalibration of the estimated scores.

Notes: AUC*, KL*, High ICI: models selected by optimizing AUC, KL divergence, or by selecting a high ICI.

Table C2: Comparison of metrics computed on the validation set for models selected based on AUC, KL divergence, or ICI, before and after recalibration. Standard errors are provided in parentheses.

| Dataset | Optim. | None | | | Platt | | | Beta | | | Isotonic | | |
|---|---|---|---|---|---|---|---|---|---|---|---|---|---|
| | | BS | ICI | KL | BS | ICI | KL | BS | ICI | KL | BS | ICI | KL |
| abalone | AUC* | 0.214 | 0.069 | 0.397 | 0.209 | 0.044 | 0.411 | 0.208 | 0.042 | 0.492 | 0.206 | 0.031 | 1.169 |
| | KL* | 0.210 | 0.057 | 0.320 | 0.208 | 0.052 | 0.475 | 0.208 | 0.048 | 0.453 | 0.205 | 0.037 | 1.239 |
| | High ICI | 0.258 | 0.180 | 4.513 | 0.219 | 0.075 | 0.383 | 0.216 | 0.057 | 0.074 | 0.214 | 0.033 | 0.770 |
| adult | AUC* | 0.090 | 0.008 | 0.325 | 0.092 | 0.039 | 0.461 | 0.090 | 0.008 | 0.328 | 0.090 | 0.008 | 0.639 |
| | KL* | 0.102 | 0.036 | 0.092 | 0.101 | 0.039 | 0.301 | 0.100 | 0.015 | 0.288 | 0.100 | 0.008 | 0.680 |
| | High ICI | 0.100 | 0.045 | 0.641 | 0.102 | 0.061 | 1.475 | 0.096 | 0.012 | 0.295 | 0.096 | 0.011 | 0.514 |
| bank | AUC* | 0.062 | 0.017 | 0.485 | 0.066 | 0.047 | 0.650 | 0.062 | 0.009 | 0.489 | 0.062 | 0.009 | 0.594 |
| | KL* | 0.070 | 0.039 | 0.062 | 0.071 | 0.037 | 0.441 | 0.069 | 0.010 | 0.315 | 0.069 | 0.004 | 0.734 |
| | High ICI | 0.068 | 0.040 | 0.437 | 0.070 | 0.040 | 0.496 | 0.067 | 0.012 | 0.453 | 0.067 | 0.005 | 0.546 |
| default | AUC* | 0.128 | 0.025 | 0.353 | 0.129 | 0.036 | 0.799 | 0.128 | 0.018 | 0.275 | 0.128 | 0.014 | 0.581 |
| | KL* | 0.129 | 0.009 | 0.349 | 0.130 | 0.027 | 0.698 | 0.129 | 0.016 | 0.234 | 0.129 | 0.014 | 0.774 |
| | High ICI | 0.142 | 0.095 | 1.498 | 0.133 | 0.036 | 1.317 | 0.131 | 0.015 | 0.660 | 0.131 | 0.013 | 0.892 |
| drybean | AUC* | 0.029 | 0.011 | 0.687 | 0.031 | 0.025 | 0.871 | 0.028 | 0.008 | 0.650 | 0.029 | 0.009 | 0.870 |
| | KL* | 0.036 | 0.042 | 0.370 | 0.040 | 0.039 | 0.843 | 0.037 | 0.030 | 0.680 | 0.034 | 0.019 | 0.731 |
| | High ICI | 0.036 | 0.070 | 2.149 | 0.034 | 0.026 | 0.832 | 0.032 | 0.021 | 0.742 | 0.031 | 0.012 | 0.771 |
| coupon | AUC* | 0.158 | 0.041 | 1.625 | 0.158 | 0.038 | 0.879 | 0.157 | 0.028 | 0.540 | 0.158 | 0.025 | 1.137 |
| | KL* | 0.192 | 0.038 | 0.048 | 0.190 | 0.025 | 0.194 | 0.190 | 0.021 | 0.132 | 0.191 | 0.022 | 0.659 |
| | High ICI | 0.162 | 0.075 | 2.354 | 0.159 | 0.054 | 1.007 | 0.157 | 0.028 | 0.522 | 0.157 | 0.023 | 1.169 |
| mushroom | AUC* | 0.000 | 0.003 | 1.399 | 0.000 | 0.003 | 1.399 | 0.000 | 0.002 | 1.399 | 0.000 | 0.003 | 1.399 |
| | KL* | 0.016 | 0.063 | 0.616 | 0.010 | 0.020 | 1.291 | 0.010 | 0.019 | 1.284 | 0.006 | 0.003 | 1.332 |
| | High ICI | 0.010 | 0.038 | 0.750 | 0.006 | 0.022 | 1.315 | 0.006 | 0.018 | 1.315 | 0.001 | 0.003 | 1.399 |
| occupancy | AUC* | 0.007 | 0.005 | 1.064 | 0.006 | 0.006 | 1.175 | 0.007 | 0.006 | 1.081 | 0.007 | 0.006 | 1.069 |
| | KL* | 0.009 | 0.044 | 0.864 | 0.007 | 0.006 | 1.136 | 0.008 | 0.006 | 1.109 | 0.008 | 0.006 | 1.055 |
| | High ICI | 0.009 | 0.033 | 0.919 | 0.007 | 0.006 | 1.157 | 0.008 | 0.007 | 1.115 | 0.008 | 0.008 | 1.052 |
| winequality | AUC* | 0.153 | 0.110 | 4.927 | 0.143 | 0.047 | 1.862 | 0.137 | 0.017 | 1.090 | 0.136 | 0.027 | 1.872 |
| | KL* | 0.170 | 0.057 | 0.118 | 0.166 | 0.040 | 0.418 | 0.166 | 0.026 | 0.365 | 0.166 | 0.018 | 1.009 |
| | High ICI | 0.153 | 0.110 | 4.927 | 0.143 | 0.047 | 1.862 | 0.137 | 0.017 | 1.090 | 0.136 | 0.027 | 1.872 |
| spambase | AUC* | 0.032 | 0.011 | 0.844 | 0.035 | 0.023 | 1.141 | 0.032 | 0.014 | 0.708 | 0.034 | 0.007 | 1.109 |
| | KL* | 0.053 | 0.055 | 0.260 | 0.049 | 0.013 | 0.560 | 0.049 | 0.013 | 0.437 | 0.050 | 0.011 | 0.858 |
| | High ICI | 0.040 | 0.035 | 0.432 | 0.041 | 0.022 | 0.865 | 0.039 | 0.015 | 0.677 | 0.040 | 0.012 | 0.975 |

Notes: AUC*, KL*, High ICI: models selected by optimizing AUC, KL divergence, or by selecting a high ICI.

