# OpenReview forum: "Post-Calibration Techniques: Balancing Calibration and Score Distribution Alignment"
_NeurIPS.cc/2024/Workshop/BDU — NeurIPS BDU Workshop 2024 Poster_

### Official Review · Reviewer_i4gQ · 2024-09-20
**Review for Post-Calibration Techniques: Balancing Calibration and Score Distribution Alignment**

**Rating:** 5
**Confidence:** 3

**Review:**

1. Summary of the Paper
The paper presents an exploration of post-processing calibration techniques, specifically Platt scaling and isotonic regression, and their impact on both model calibration and the alignment between score distributions and true event distributions. The authors employ simulated and real-world datasets to demonstrate that while these techniques improve traditional calibration metrics (like Brier Score or Integrated Calibration Index), they often lead to divergence between score distributions and actual event probabilities. The work primarily focuses on XGBoost binary classifiers.
2. Strengths
• The paper addresses a significant problem in model calibration, highlighting the potential risks of relying solely on calibration metrics without considering score distribution alignment.
• The experiments are well-designed, comparing different calibration techniques and showcasing their strengths and limitations through rigorous analysis using both simulated and real-world data.
• The paper’s discussion on the limitations of current calibration methods and the proposal to optimize model parameters based on Kullback-Leibler (KL) divergence offers a fresh perspective on the topic.
3. Weaknesses
• The explanation of score heterogeneity in the paper is insufficiently formalized and lacks a coherent structure, making it challenging to understand its exact meaning and significance in the study. Furthermore, the specific definition and implications of score heterogeneity are underdeveloped, resulting in an unclear presentation of its role.
• The paper lacks deeper theoretical grounding for the observed effects of calibration techniques on score heterogeneity. While empirical results are provided, the reasons behind the divergence between calibration and score distribution alignment need more rigorous explanation.
• The work only evaluates two common calibration methods (Platt scaling and isotonic regression). The inclusion of more modern techniques (e.g., beta calibration or spline-based methods) could have enriched the comparison.
4. Minor Comments
• The sentence “Isotonic regression, like Platt scaling, assumes …” appears to contain an error.
• Figures and tables could be enhanced to make the results more interpretable, especially for those unfamiliar with calibration and distribution alignment.

---

### Official Review · Reviewer_r72s · 2024-09-24
**The paper discusses post-processing calibration methods Platt scaling and isotonic regression to align the true and predicted probability distribution of data in a binary scoring classifier model. It compares the performance of an XGBoost model before and after applying these calibration techniques using simulated data with known true probabilities and real-world datasets with prior event distribution knowledge.**

**Rating:** 5
**Confidence:** 4

**Review:**

Summary

The paper discusses post-processing calibration methods Platt scaling and isotonic regression to align the true and predicted probability distribution of data in a binary scoring classifier model. It compares the performance of an XGBoost model before and after applying these calibration techniques using simulated data with known true probabilities and real-world datasets with prior event distribution knowledge.

Strengths

1 - Good introduction, I like how the first paragraph is structured.

2 - Isotonic Regression - I like the fact that the paper didn't use acronyms and specified the algorithm name (Pool-Adjacent-Violators Algorithm)

3 - Score Heterogeneity - I like how the paper frames and describes the problem of calibration metrics not being able to fully capture the discrepancies.

4 - Kullback-Leibler divergence - The paragraph is well explained and gives an adequate amount of information needed.

5 - Simulated Data - Good experiment design. I like that the calibration technique is applied both before and after comparing the models.

Weakness

Major Weakness

1 - Section 2.1 - Insufficient information about BS. There's more explanation about ICI only and not BS.

2 - Section 2.1 - Insufficient explanation about the 45-degree diagonal line. I was not able to understand the relevance of this and why specifically this.

3 - Section 2.2 - What are mu and s? There's no clear explanation of these 2 terms. Is 's' the same as mentioned in 1st paragraph of section 2? Only the last line of this paragraph is relevant and the rest should have been explained in a better way.

4 - Section 2.2 - What is beta? Why is beta_1<=...<=beta_n? I was unable to understand the equation because of the beta term, a better explanation should have been provided. Again, only the last line in this paragraph is what I feel is relevant.

5 - Section 4 - This section does not provide any explanation of the results obtained. No explanation has been provided on how to analyze the graphs/plots provided. There should have been some information about how to interpret the graphs/plots and how to infer from them.

6 - Appendix - A lot of graphs/plots are provided for various noise levels without any explanation about them and how to infer a conclusion from them. I was not able to analyze the graphs at all as it is very difficult to infer from them without any explanation or direction. There are also a lot of tables that lack clear explanations on what to infer from them.

7 - No validation of results - It doesn't mention any means to validate the results provided.

Minor Weakness

1 - Section 1 - Citation issue in introduction 2nd paragraph. The author's name should be mentioned.

2 - Section 3 - Kullback-Leibler divergence - citation format issue. The author's name should be mentioned.

3 - Section 3 - Bayesian Framework - Improper citation format - In this paragraph the paper by Fernandes et al has been cited twice, rephrase the sentence.

4 - Section 3 - Bayesian Framework - The formulae provided are not necessary, since this is not the method being followed. Potentially save some space.

5 - Section 3 - Calibration techniques - Improper citation format for [9]

6 - Section 4.1 - Simulated Data - Improper citation format for both [9] and [25]

7 - Section 4.1 - Simulated Data - This paragraph mentions optimizing based on 3 criteria in this sentence:- "We select the model’s hyperparameters (number of boosting iterations and maximum tree depth) to optimize three different criteria on the validation set: maximizing AUC (AUC*), minimizing KL divergence (KL*), or, for illustrative purposes, producing a model that is poorly calibrated based on the ICI metric (High ICI)." Even though this is grammatically correct it could be clear in its structure. Use 'and' instead of 'or'.

Remarks

If there is not enough space to briefly describe something, I would suggest citing it in the main paper body and adding a section in the appendix, that explains it clearly. Otherwise, an incomplete amount of information creates more confusion.

Conclusion

Overall, the paper provides valuable information about post-processing calibration methods for aligning probability distributions in binary classifiers. The introduction and problem framing is strong, and the experimental design effectively evaluates calibration techniques before and after model application. However, the paper lacks detailed explanations for key metrics and concepts, which hampers reader understandability. Additionally, the experimental results are presented without adequate analysis or interpretation, leaving the findings underexplored and difficult to validate. The paper would benefit from clearer explanations of the methodologies, better contextualization of results, and a more thorough discussion of their implications.
I recommend rejecting the paper in its current form and suggest that the authors provide more detailed explanations, particularly regarding the results, to improve the clarity and impact of their work.

---

### Decision · Program_Chairs · 2024-10-09

**Decision:**

Accept (Poster)

**Comment:**

This paper has borderline scores. The reviewers are mainly worried about presentation quality and not enough aspects being explained. At least part of this stems from the short four-page workshop format, and possibly missing information in the appendix that could be added in the camera-ready. I think most of this can be addressed in the camera-ready, and therefore recommend acceptance.